# Cryo-EM reveals the molecular basis of laminin polymerization and LN-lamininopathies

Arkadiusz W. Kulczyk [1] ✉, Karen K. McKee [2], Ximo Zhang[3], Iwona Bizukojc[3,4], Ying Q. Yu[3] & Peter D. Yurchenco[2]

Laminin polymerization is the major step in basement membranes assembly. Its failures cause laminin N-terminal domain lamininopathies including Pierson syndrome. We have employed cryo-electron microscopy to determine a 3.7 Å structure of the trimeric laminin polymer node containing α1, β1 and γ1 subunits. The structure reveals the molecular basis of calcium-dependent formation of laminin lattice, and provides insights into polymerization defects manifesting in human disease.

Laminins (Lm) are key components of basement membranes (BMs). They form a planar lattice on cell surfaces. The Lm polymer node, which constitutes the repeating unit of Lm lattice, is a flexible N-glycosylated heterotrimer consisting of α, β and γ subunits that join together through their N-terminal short arms[1–3], while the C-terminal long arms form a coiled-coil, from which extends a cluster of globular domains that bind to cell surface receptors. (Fig. 1a). In mammals, there are five Lm α, four β, three γ, and two splice variants (α3 A, α3B), which assemble into functional trimers in at least fifteen combinations[4]. Crystal structures of N-terminal fragments from Lm α5 (a homolog of α1)[5], β1[5], and γ1[6] revealed elongated molecules with LN domains consisting of the seven- or eight-stranded antiparallel β-sheets at the N-terminus, followed by a rod-like tandem of random-coil LE domains containing the EGF-like fold[7]. Genetic loss of Lm subunits results in a group of disorders[8], which we define as Lm N-terminal domain lamininopathies (LN-lamininopathies), including failures of early differentiation and organogenesis (α1, β1, γ1, α5), and diseases manifesting in kidney/eye (β2) or muscle/peripheral nerve/brain (α2). Polymerization failures occur in a subset of patients with Pierson syndrome and Lm α2-related dystrophy[9]. In addition, disruption of Lm polymer impedes cancer metastases[10].

Lm polymerization, first described by Yurchenco and colleagues[11], is a nucleation-propagation assembly with an initial calcium-independent oligomerization step involving β and γ subunits, followed by a calcium-dependent aggregation step, in which α associates with the β-γ dimer (Fig. 1a). The polymer, which is stabilized by a network of reversible bonds[1,12], has the architecture visualized as a mesh of interconnecting struts in platinum-carbon replicas by transmission electron microscopy[12]. The X-ray structures of the LN-LE fragments from Lm α5[5], β1[5] and γ1[6] revealed foot-like shaped molecules with regions resembling the heel and the toe. These two regions contain residues required for the Lm polymer node assembly[13]. These residues are located on one side of the molecules opposite of the side containing N-glycans[5,6]. A size-exclusion chromatography (SEC) analysis of native and altered Lm variants was employed to show that the β-γ dimerization step is of low affinity ($K_D = 20–22$ μM), whereas the α to β-γ trimerization step is of higher affinity ($K_D = 2$ μM)[13]. During BM assembly, the Lm polymer forms a molecular layer anchored to integrin and α-dystroglycan receptors through the C-terminal globural domains of Lm α. The Lm polymer is further attached to the collagen-IV network, largely through nidogen-1 and possibly through heparan sulfates[3]. In addition, agrin binds to the Lm coiled-coil, and perlecan binds to nidogen, with both agrin and perlecan also associating with α-dystroglycan to stabilize the BM[3].

Understanding the process of Lm polymerization requires the knowledge of Lm polymer node structure. Due to its intrinsic flexibility, a trimeric Lm polymer node has been elusive to structure determination. Consequently, the mechanisms for Lm polymerization, and molecular basis underlying LN-lamininopathies are poorly understood.

## Results and discussion
### Cryo-EM structure of the Lm polymer node
Recent advances in cryo-EM provide unprecedented insight into structures of dynamic macromolecular complexes[14,15]. We employed

[1]Institute for Quantitative Biomedicine, Department of Biochemistry and Microbiology, Rutgers University, Piscataway, NJ 08854, USA. [2]Department of Pathology and Laboratory Medicine, Rutgers University - Robert Wood Johnson Medical School, Piscataway, NJ 08854, USA. [3]Waters Corporation, Milford, MA 01757, USA. [4]Cryo-EMcorp, Bridgewater, NJ 08807, USA. ✉e-mail: arek.kulczyk@rutgers.edu

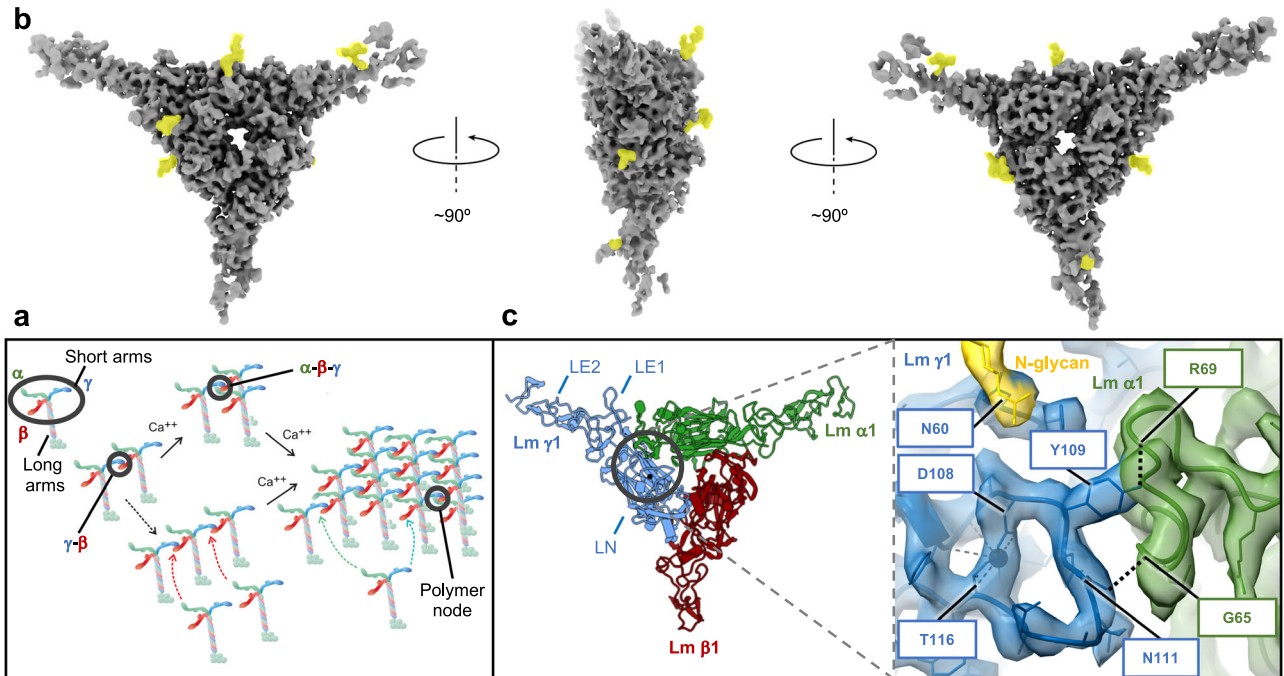

**Fig. 1 | Cryo-EM structure and mechanisms of Lm polymerization. a** A model for Lm assembly and polymerization. Calcium independent formation of β1-γ1 dimers is followed by a calcium dependent association of α1, and the extension of Lm matrix. Lm short and long arms, and one of the trimeric Lm polymer nodes formed by interacting short arms from Lm α1, β1 and γ1 are highlighted in black. **b** Different views of the cryo-EM map. The N-glycans are colored in yellow. **c** Structural details of the loops stabilizing the α1-γ1 interface. Lm α1, β1 and γ1 are shown in green, red and blue, respectively. The loop from γ1 contains residues (D108 and T116) coordinating a calcium ion. The calcium ion is shown as a black sphere. The α1-γ1 interface is stabilized by a network of hydrogen bonds indicated as black dotted lines.

cryo-EM to determine a 3.7 Å structure of Lm complex consisting of 56 kDa α1, 64 kDa β1, and 52 kDa γ1, assembled into a functional trimer (Fig. 1b). Details of the SPA workflow employed for the cryo-EM structure calculations are displayed in Supplementary Fig. 1a–j, Supplementary Fig. 3–5, and in Supplementary Table 1. The discovery of the stabilizing mutation in γ1 subunit and previous negative-stain EM analysis provided the foundation for the cryo-EM study[13]. Because of the time- and concentration-dependent instability of Lm trimers, samples were vitrifying within minutes after elution from the GF column at the concentration favoring timer formation (Supplementary Fig. 1a–d). We employed a combination of homology modeling implemented in Phenix, de novo model building in COOT, and deep learning algorithms implemented in AlphaFold2 to obtain an initial model of the complex (Supplementary Fig. 6). Despite the structural homology, individual Lm subunits adopt substantially different conformations in the complex, allowing for unambiguous assignment of α1, β1 and γ1 to the Coulomb map of the trimeric Lm polymer node (Supplementary Fig. 7, 8). To confirm the assignment of subunits, we exploited differences in their N-glycosylation patterns. The cryo-EM map showed the presence of extended densities attached to its surface (Fig. 1b, c and Supplementary Fig. 2). The MS glycopeptide analysis revealed eight unique N-glycosylation sites (Supplementary Fig. 9), which colocalize with protrusions in the 3D Coulomb map. In addition, the Q-score analysis of all possible model vs. map combinations supports the assignment of subunits (Supplementary Table 2).

The asymmetric cryo-EM structure of the Lm polymer node resembles a triskelion with centrally located LN domains and three rod-like structures projecting outwards, each containing one or two LE domains (Fig. 1a, c). The first LE domains pack against the outer surfaces of β-sheets from LN domains, creating a bend ranging from 110 degrees to 130 degrees between LN domains and LE rods. The Principal Components Analysis revealed the planar and rotational mobility of LE rods (Supplementary Fig. 10 and Supplementary Movie 1). The LN domain in γ1 uniquely contains a calcium-binding site (Fig. 1c, Supplementary Fig. 2c and Supplementary Fig. 18a, b). The complex is stabilized by a network of disulfide and hydrogen bonds, and electrostatic and hydrophobic interactions (Supplementary Fig. 11). The inter-subunit interfaces (Fig. 2) are formed by two sets of interacting regions (Supplementary Fig. 12). The first set involves loops connecting β1 and β2 strands in β-sheets from neighboring subunits, whereas the second set includes loops linking strands β7 and β8 from one subunit, and the N-terminal region along with one of the loops from the LE1 domain in the neighboring subunit.

## Mutational analysis of inter-subunit interfaces

Site-directed mutagenesis of inter-subunit interfaces, followed by SEC analysis of Lm oligomers reconstituted with genetically-altered subunits, identified residues affecting trimer formation, for instance R208/A and R208/A-D218/R in β1 (Supplementary Fig. 13). In addition, fifteen previously reported amino acid alternations resulting in disruption of Lm trimers[13] can be mapped to the inter-subunit binding interfaces (Supplementary Fig. 14). For example, the analysis of the cryo-EM structure provides the detailed mechanistic explanation why the following mutations disrupt the trimeric structure of the Lm polymer node: in α1 (Y128R, E203R, R263D, Supplementary Fig. 15), β1 (S68R, S200R, E204R, R208E, Supplementary Fig. 16), and γ1 (Y147R, S213R, D261R, Supplementary Fig. 17). In the majority of cases the aforementioned amino-acid alternations disrupt the network of hydrogen bonds stabilizing the neighboring subunits within the trimer. The γ1D266R mutation stabilizing the complex is located at the γ1-β1 interface (Supplementary Fig. 14).

## Molecular mechanism for Lm polymerization

The interface involving α1 and γ1 differs from other inter-subunit interfaces, as it is mainly electrostatic in nature (Supplementary Fig. 11b) and it involves a calcium-binding site from γ1[6] (Fig. 1c,

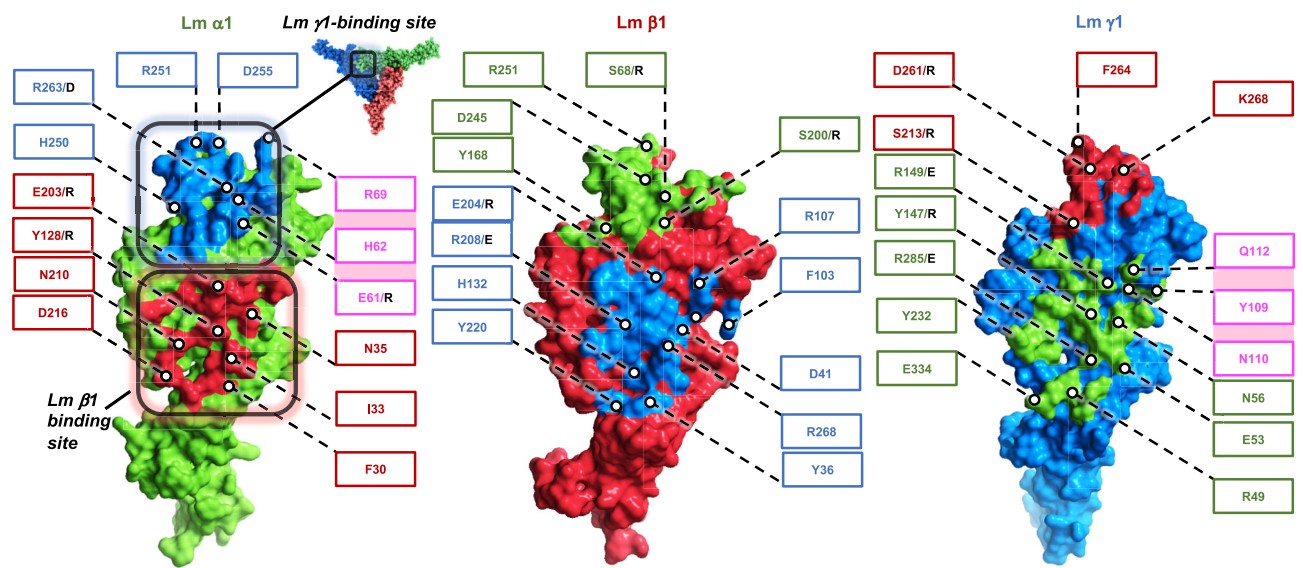

**Fig. 2 | Inter-subunit interfaces stabilizing the Lm polymer node.** Lm α1 and inter-subunit interfaces formed by α1 with β1 and γ1 are shown in green. Lm β1 and γ1 subunits and their correspondent inter-subunit interfaces are displayed in red and blue, respectively. Previously described mutations resulting in disruption of the trimeric structure of Lm complex are indicated by black letters. Some of the residues involved in calcium-dependent association of β1-γ1 dimers with α1 are indicated by pink boxes. The interface involving α1 and γ1 is larger than interfaces formed by α1-β1 and β1-γ1 (1372.2 Å², 1227.4 Å² and 1138 Å², respectively).

Supplementary Fig. 2c and Supplementary Fig. 18a, b), the latter fact with implications for the mechanism of sequential assembly of LM trimer. The cryo-EM structure reveals that the α1-γ1 interface involves a loop spanning residues K58-Q72 in α1, and another loop from γ1 containing residues L106-T116 (Fig. 1c and Supplementary Fig. 18a). The loop from γ1 consists of amino acids critical for coordination of a calcium ion, including D108 and T116[6]. We propose that in the absence of a calcium ion the loop in γ1 is not structured, hence the α1-γ1 interface cannot be formed, explaining the calcium dependence for the assembly of a trimeric Lm node.

## Molecular basis underlying Pierson syndrome

Pierson syndrome is an autosomal recessive disorder accompanied by neurological abnormalities, frequently leading to irreversible kidney failure. On the molecular level, a subset of Pierson syndrome results from mutations in the human LAMB2 gene encoding the LN domain of Lm β2 chain[9]. The LN domains from β1 and β2 share 72% sequence identity, with all residues implicated in the disease conserved in both variants (Supplementary Fig. 19). The Pierson LN mutations include S68R, L127P, R234W and R234Q (Fig. 3). S68 is located in one of Lm β loops spanning residues I66-K75, which packs against the inner face of the β-sheet in α1. A substitution of neutral S68 with a larger and positively charged arginine disrupts the network of neighboring hydrogen bonds stabilizing the β1-α1 interface (Fig. 3b). L127 is located in a hydrophobic core of the eight-stranded β-sheet. A substitution of L127 with proline destabilizes the hydrophobic core of β1, likely affecting its folding and interactions with the neighboring γ1 (Fig. 3c). R234W and R234Q are frequent mutations producing severe phenotypes[16]. R234W leads to a significant decrease of Lm expression, while R234Q reduces extracellular Lm secretion. R234 is positioned on the outer surface of the β-sheet in the location adjacent to one of the two invariant N-glycosylation sites, N120 (Fig. 3d). Substitution of a positively charged R234 with an indole ring of tryptophan or a neutral glutamine likely affects the N-glycosylation and folding of β1.

In summary, we determined a cryo-EM structure of the Lm polymer node. The structure reveals fundamental molecular mechanisms governing formation of extracellular Lm matrix. Importantly, the structure provides insight into the molecular basis underlying Pierson syndrome and other related LN-lamininopathies. The structure offers to facilitate rational drug design aiming in the treatment of Lm deficiencies, and can foster development of biomimetic BMs for tissue implants.

## Methods

### Protein Expression and Purification

The recombinant proteins were purified from human embryonic kidney cells (HEK293, ATCC crl-1573 tm) stably expressing αLNLEa N-FLAG, β1LNLEa N-HA and c-FLAG or γLNNd D266R c-FLAG[13]. The HEK293 cells were cultured in DMEM (Invitrogen, 11995-081) supplemented with 10% Fetal Bovine Serum (Atlanta Biological, S11150), 200mM L-Glutamine and Penicillin-Streptomycin (1,000 u/ml Penicillin and 1,000 µg/ml Streptomycin, Invitrogen, 15140122) as well as 1µg/ml Puromycin (Invitrogen, J67236XF), 100ug/ml Zeocin (Invitrogen, R25001) or 500 µg/ml G418 (Invitrogen, 11811023), respectively. All proteins were purified from media using anti-FLAG M2-agarose (Sigma, A2220), concentrated in an Amicon Ultra-15 filter (30 K MWCO), (Millipore UFC903024), and dialyzed in TBS50 (90 mM NaCl, 50 mM Tris pH 7.4, 0.125 mM EDTA). Protein concentrations were determined by absorbance at 280 nm. For FPLC purification of the trimer complex, 300 µg of each proteins was mixed and concentrated in Amicon Ultra 0.5 ml (Millipore, 10 K), (Millipore UFC501034) to 150 µl (final concentration 6 mg/ml) and incubated with 1 mM calcium at 27 C for 1 h. The trimer mix was injected into a Superdex 200 Increase 10/300 GL (GE Healthcare, 28990944) column connected to an AKTA FPLC system (Pharmacia/GE Healthcare) with a flow rate of 1.0 ml/min at room temperature. The peak trimer fraction (0.5 ml, 0.4 mg/ml to 0.6 mg/ml) was diluted to the desired concentration in 20 mM HEPES, 150 mM NaCl, 20 mM imidazole, 2 mM CaCl₂, and 0.02% NaN₃, pH 7.5, and used for preparation of cryo-EM grids.

### Site-directed mutagenesis of the inter-subunit interfaces in the Lm polymer node

The β1 variants: R208/A and R208/A-D218/R were expressed and purified, and their oligomerization pattern was evaluated by size-exclusion chromatography (SEC) with a Superdex 200 Increase 10/

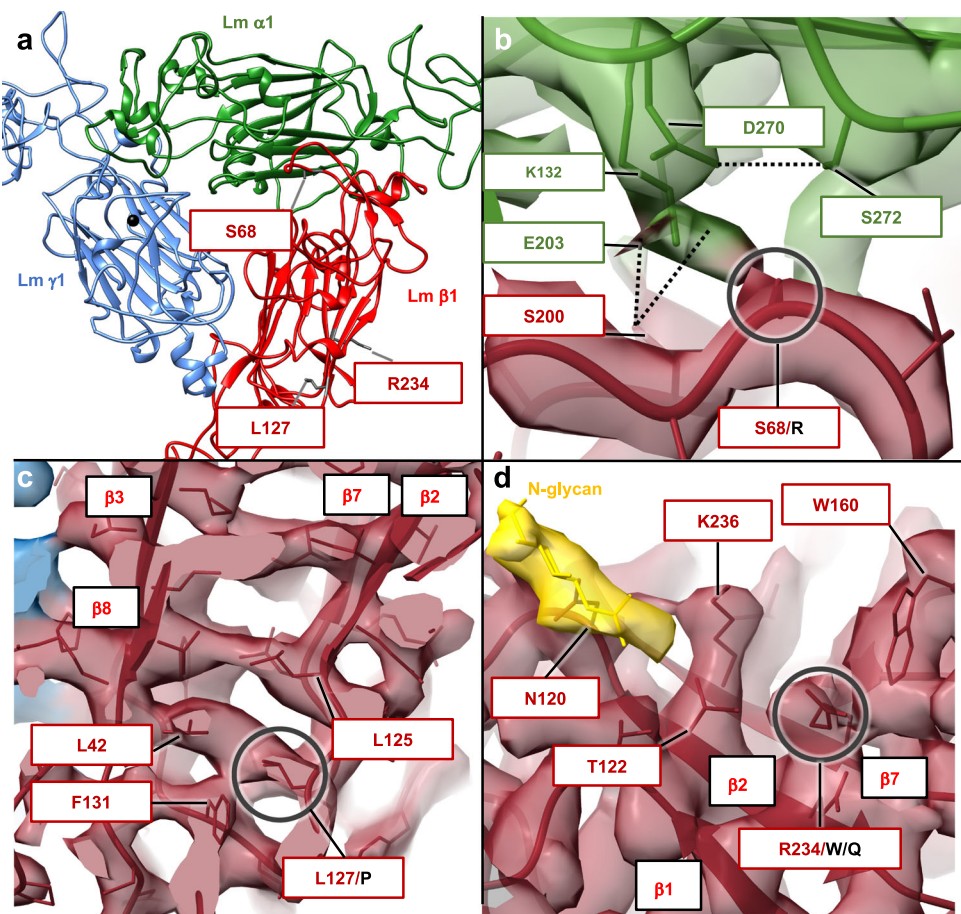

**Fig. 3 | Molecular basis of Pierson syndrome. a** Mutations causing Pierson syndrome are located in β1 and indicated by red boxes in the figure. Lm α1, β1 and γ1 are shown in green, red and blue, respectively. A calcium ion is shown as a black sphere. **b** S68R destabilizes the β1-α1 interface by disrupting a network of hydrogen bonds indicated in the figure as black dotted lines. **c** L127P destabilizes the hydrophobic core of β1 formed by an inner face of the seven-stranded β-sheet. Individual strands of the β-sheet are labeled. **d** R234W and R234Q mutations likely affect N-glycosylation of the neighboring N120. In addition, W234 may form stacking interactions with W160 destabilizing the β-sheet in β1. The N-glycan is displayed in yellow.

300 GL (GE Healthcare, 28990944) column using an AKTA FPLC system (Pharmacia/GE Healthcare) at room temperature. The flow rate was 1.0 ml/min. The peak trimer fraction (0.5 ml, 0.4 mg/ml to 0.6 mg/ml) was diluted to the desired concentration in 20 mM HEPES, 150 mM NaCl, 20 mM imidazole, 2 mM CaCl$_2$, and 0.02% NaN$_3$, pH 7.5. In addition, we analyzed previously published[13] mutagenesis and SEC results of a series of α1 (Y128/R, E203/R, R263/D), β1 (S68/R, S200/R, E204/R, R208/E), and γ1 (Y147/R, S213/R, D261/R) variants. The β1 S68/R mutation was originally introduced by Purvis et al. [17].

### Glycopeptide analysis by UHPLC-MS

The analysis of glycosylation sites was performed via enzymatic digestion of the three Lm subunits followed by LC-MS analysis. Briefly, 5 µg of α1, β1 and γ1 were denatured using a surfactant, *Rapi*Gest™ SF (Waters Corporation, Milford, MA), at a concentration of 0.05% (w/v) and 1 mM dithiothreitol in 50 mM Tris-HCl buffer (pH = 8) at 80 °C for 3 min. The denatured Lm subunits were digested using a sequence-grade modified trypsin (Promega, Madison, WI) with an enzyme-to-substrate ratio of 1:10 at 37 °C for 1 h. The 2 µg of each protein digest was then loaded on a 2.1 × 150 mm ACQUITY UPLC BEH peptide C18 column (1.7 µm particle size, 300 Å pore size) at 60 °C and analyzed on a LC-TOF MS system (BioAccord™, Waters Corporation, Milford, MA) consisting of an Acquity UPLC I-class PLUS and an RDa time-of-flight mass detector. Mobile phase A and B were 0.1% formic acid in LC-MS grade water and acetonitrile, respectively. At a constant flow rate of 0.2 mL/min, a linear gradient from 98–50% mobile phase B was used

for separation. The MS settings were set as follows: mass scan range, 50–2,000 m/z; capillary voltage, 1.5 kV; cone voltage, 45 V; fragmentation cone voltage ramping, 100–120 V; and desolvation temperature, 500 °C. The acquired LC-MS data were analyzed using peptide mapping workflow within the Waters_connect informatics platform. Mass accuracy was set at 10 ppm for precursor ions and 20 ppm for fragment ions confirmations.

### Released N-glycan analysis

The N-glycans were released from individual Lm subunits and fluorescently labeled using the GlycoWork Rapifluor-MS N-glycan labeling kit (Waters, Milford, MA) following a previously published method. The labeled N-glycans were analyzed using a 2.1 × 150 mm ACQUITY UPLC BEH amide glycan column (1.7 µm particle size, 130 Å pore size) on the LC-MS system. The column temperature was 60 °C. Mobile phase A was 50 mM ammonium formate in LC-MS grade water (pH = 4.4), while mobile phase B was 100% LC-MS grade acetonitrile. The separation was carried out at a flow rate of 0.4 mL/min with the following gradient: 75–54% B in 35 min, 54–20% B in 35.5 min, 20% B in 39.5 min, 20–75% B in 43.1 min, and 75% B in 55 min. The same MS settings from peptide mapping were applied in released glycan analysis except using a fragmentation cone voltage of 70–90 V and a desolvation temperature of 300 °C. The Released Glycan Workflow in Waters_connect informatics platform was used for data processing and reporting. Mass accuracy was set at 10 ppm for precursor ions and 20 ppm for fragment ions.

## Cryo-EM sample preparation, screening and data collection

Samples of the Lm polymer node were prepared by mixing three individual subunits (α1, β1, γ1) at the equimolar ratio. The γ1 subunit contained an amino-acid substitution (γ1D266/R), which enhanced stability of the trimer. The stability of Lm111D266R trimers is time- and concentration-depended, hence samples for cryo-EM were frozen within 15–30 min after elution from the gel filtration column at the concentration of 90 μg/ml. Samples were adsorbed onto freshly glow-discharged 300 mesh UltrAuFoil R1.2/1.3 (Millipore Sigma, TEM-Q350AR13A) or gold Quantifoil R2/1 (SPI Supplies, 4330G-CF) grids with PELCO easiGlow glow discharger and flash frozen in liquid ethane using either Vitrobot Mark IV dual-blotting plunger or Leica GP Climate controlled sample plunger with a controlled temperature and humidity. The grids were extensively screened in a total of sixteen multi-day data collection sessions at no stage tilt or at 40[0] stage tilt with a 200 kV Thermo Fisher Scientific Talos Arctica electron microscope equipped with a Gatan BioQuatum energy filter and Gatan K2 Summit direct electron detector at the Center for Integrative Proteomics Research at Rutgers University using the software SerialEM 4.0[18] or EPU 2 for automated data collection. Typical acquisition parameters were as follows: dose rate of 4.88 e⁻/px/s in a counting mode, magnification of 130,000 times corresponding to the pixel size of 1.037 Å/px with a defocus range −0.5 to −2.5 μm. We collected 32 frames with 250 ms per frame exposure, a total exposure of 8 s and an accumulated dose of 39.04 e⁻. The screening procedure also involved pre-processing, 2D classification and class-averaging of the acquired data set in cryoSPARC v3. Cryo-EM grids, in which the majority of particles yielded 2D classes displaying different views of a trimeric Lm complex were shipped to the Pacific Northwest Center for Cryo-EM (PNCC) for data collection with 300 kV Thermo Fisher Scientific Titan Krios equipped with an energy filter and Gatan K3 direct electron detector. The data were acquired using software SerialEM 4.0[18]. At the PNCC, we collected 44,743 movies during 2 individual sessions. During the first session (data set #1) 20,618 movies were recorded at no stage tilt using the following acquisition parameters: dose rate of 17.5 e⁻/px/s in a super-resolution mode, magnification of 130,000 times corresponding to the pixel size of 0.324 Å/px with a defocus range −0.8 to −2.2 μm. We collected 29 frames with 0.78 electrons per pixel per frame, a total exposure of 1.255 s and an accumulated dose of 52.3 e⁻/A². In the second session (data set #2), 24,125 movies were collected at both, no stage tilt and at 45[0] stage tilt, using the following acquisition parameters: dose rate of 16 e⁻/px/s in a super-resolution mode, magnification of 130,000 times corresponding to the pixel size of 0.324 Å/px with a defocus range −0.8 to −2.2 μm. We collected 37 frames with 0.6 electrons per pixel per frame, a total exposure of 1.4 s and an accumulated dose of 52 e⁻/A².

## Structure determination

A structure of the trimeric Lm complex was calculated using a combination of cryoSPARC v3[19], Scipion-3[20], and Relion-3[21]. The data acquired during sessions 1 and 2 (i.e. at no stage tilt and at 45[0] stage tilt) were combined for calculations of the final structure. Movies were patch motion- and patch CTF-corrected. Following inspection of pre-processed averaged micrographs, 6816/44743 micrographs were discarded from the data set. We manually picked 500 particles from the selected micrographs to create 2D templates for automated particle picking. The template picker automatically selected 12,556,133 particle images. Selected particles were inspected and extracted from averaged micrographs with a box size of 900 pixels, and then down-sampled by a factor of 3 resulting with a pixel size of 0.972 Å/px. Particle stacks were subsequently subjected to 6 rounds of 2D classification and averaging. During the 2D analysis, artifacts and particles not converging into stable classes were removed from the data set. The resultant set of 1,068,172 particle images was used for ab-initio modeling. To assess the heterogeneity in the sample, we carried out ab-initio modeling with 1 to 5 classes, followed by multiple rounds of heterogenous refinements during which additional 943,161 particle images were discarded. The resultant maps were used as inputs for multiple rounds of homogenous and non-uniform refinements. All 3D maps had C1 symmetry. The final map calculated with 125,011 particle images had an estimated resolution of 3.7 Å, as calculated using the gold standard Fourier Shell Correlation (FSC) method according to the 0.143 criterion. We employed a deep-learning algorithm, DeepRes[22], available in Scipion-3[20,23] to calculate local resolutions, and LocalDeblur[24] to locally sharpen the map according to the local resolution estimates. The Principal Components Analysis (PCA) implemented in cryoSPARC 3D Variability script was performed with a set of 213,694 particle images. This set was obtained after initial 3D hetero-refinement, which removed artifacts and Lm monomers from the data. We calculated three eigenvectors of the 3D covariance reflecting possible molecular motions in the Lm polymer node structure.

In attempt to refine the map further, we performed local refinements with image subtraction in cryoSPARC v3. UCSF Chimera[25] and UCSF ChimeraX[26] were used to segment the map into: (i) fragments corresponding to individual Lm subunits, or (ii) the central core containing a trimeric LN complex, and three rod-like tandems of LE domains from individual subunits. All segments were then used to create masks for focused refinements. Because the above approach didn't improve the resolution of the map, we attempted to re-process the data in RELION-3[21] de novo or by transferring coordinates of particles, which yielded a 3.7 Å map, from cryoSPARC v3 to RELION-3 using PyEM and in-house scripts[27,28], and also in Scipion-3. Relion-3 and Scipion-3[20] calculations didn't improve the resolution of the structure.

## Model building

The initial model of the trimeric Lm complex was obtained using a combination of Phenix 1.18[29], COOT 0.9[30], UCSF Chimera[25], and Colab-Fold with AlphaFold2 and the sequence search module MMseq2[31]. In brief, atomic coordinates of previously determined X-ray structures of α5 (PDB ID: 2Y38 [https://doi.org/10.2210/pdb2Y38/pdb][5], β1 (PDB ID: 4AQS [https://doi.org/10.2210/pdb4AQS/pdb][6] and γ1 (PDB ID: 4AQT [https://doi.org/10.2210/pdb4AQT/pdb])[6] were docked into the Coulomb density map in UCSF Chimera. The model was inspected and adjusted in COOT 0.9. To facilitate the convergence of the initial model, we also employed AlphaFold2[32]. AlphaFold2 was first used to build a model of α1 for which a crystal structure is not available, and then a model of the trimeric Lm complex. Protein regions missing from the crystal structures were either built de novo in COOT 0.9 or adopted from AlphaFold2 model, and then adjusted in COOT 0.9. To ensure the correct assignment of Lm chains to each individual segment in the map, we have initially inspected all possible chain vs. map segment combinations. A visual inspection, as well as quantitative analysis with Phenix[30] 1.18 and MapQ[33] made it apparent that α1, β1 and γ1 adopt substantially different conformations within the trimeric Lm complex, thus allowing for unambiguous assignment of subunits. In particular, the loop regions involved in the inter-subunit interactions have different lengths and adopt different 3D structures. In addition, the cryo-EM map of the Lm polymer node revealed eight extended densities representing N-glycans. The unique N-glycosylation pattern of individual Lm subunits was identified with UHPLC-MS by employing protein samples analogous to these used for cryo-EM structure determination. The molecular model of the Lm polymer node revealed that the densities representing N-glycans could be linked to eight uniquely positioned asparagine residues within the trimer, which confirmed the correctness of the initial model. Such scenario would not be possible for any other arrangement of Lm subunits within the trimer. Also, to validate the model, we calculated the Q-scores for all possible model and map combinations. The calculations confirmed the correctness of the assignment. The model of the Lm polymer node was then iteratively refined using a combination of COOT 0.9[30] and Phenix 1.18[29] with

imposed NCS, secondary structure and Ramachandran restrains. We systematically improved the quality of the model by addressing specific issues, such us interatomic clashes, violations of Ramachandran and other geometry restrains including errors in side chain rotamers. Each specific violation was addressed in COOT 0.9 in between subsequent Phenix 1.18 refinement runs. The final model was validated using Mol-Probity 4.02. The final model displays good validation statistics presented in Supplementary Table 1, and reveals many structural details specific for each unique Lm subunit, including a number of well-defined amino-acid side chains and inter-subunit contacts confirmed by the mutagenesis analysis. Five uniquely-positioned N-Acetylglucosamine (NAG) moieties were added to the model. Although, the densities representing N-glycans in the map reach well beyond the first NAG, we decided not to further extend the N-glycan chains in the model, as we wouldn't be able to do it with high confidence. Also, the segment representing Lm γ1 in the map displays the density representing calcium coordination, hence we added the calcium ion to the model.

### Reporting summary

Further information on research design is available in the Nature Portfolio Reporting Summary linked to this article.

## Data availability

Atomic coordinates of Lm polymer node generated in this study were deposited in the Protein Data Bank (PDB) under accession code 8DMK [https://doi.org/10.2210/pdb8DMK/pdb]. The sharpened and unsharpened cryo-EM maps were deposited in the Electron Microscopy Data Bank (EMDB) under accession code EMD-27542. Atomic coordinates of previously determined X-ray structures are available in the PDB under the following accession codes: 2Y38 (Lm α5), [https://doi.org/10.2210/pdb2Y38/pdb], 4AQS (Lm β1), [https://doi.org/10.2210/pdb4AQS/pdb], and 4AQT (Lm γ1), [https://doi.org/10.2210/pdb4AQT/pdb]. Source data are provided with this paper.

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

## Acknowledgements

We thank Donald Winkelmann (Rutgers University) for careful reading of the manuscript, and Erhard Hohenester (Imperial College London) for a useful discussion and funding support of one of the cryo-EM screening sessions. We are grateful to Janette Myers and Claudia Lopez for their assistance in data collection at the PNCC. A portion of this research was supported by NIH grant U24GM129547, and performed at the PNCC at OHSU and accessed through EMSL (grid.436923.9), a DOE Office of Science User Facility sponsored by the Office of Biological and Environmental Research. This study was supported by a NIH grant R01-DK36425 to P.D.Y, a PNCC-51312 grant to A.W.K., and a start-up grant from Rutgers University to A.W.K.

## Author contributions

A.W.K. conceptualized the cryo-EM project with the input from P.D.Y. and K.K.M., collected and analyzed cryo-EM data, calculated the structure, built and refined the molecular model, wrote the original manuscript, and revised the manuscript with input from P.D.Y and I.B. K.K.M and P.D.Y. purified and reconstituted the Lm complex for cryo-EM, conducted SEC analysis of Lm mutants, and assisted in grid freezing. X.Z., I.B. and Y.Q.Y. performed MS analysis.

## Competing interests

The authors have no conflict of interest.
