## [Peer Review File · Nature Communications]

Editorial Note: This manuscript has been previously reviewed at another journal that is not operating a transparent peer review scheme. This document only contains reviewer comments and rebuttal letters for versions considered at *Nature Communications*

REVIEWERS' COMMENTS

Reviewer #1 (Remarks to the Author):

Comments to Kulczyk et al.(NCOMMS-22-39611-T)

I have reviewed the previous version of this paper for Nature Struct Mol Biol and gave number of comments and suggestions. In this extensively revised version, the authors addressed most of my concerns and criticisms by performing additional experiments and re-analyses, adding many supplemental display items, and adding detailed description in the methods section. I am satisfied with these revisions and can confidently recommend acceptance of this paper for Nature Communications.

Jun Takagi

Cryo-EM reveals the molecular basis of laminin polymerization and LN-laminopathies.

Reviewer #1 (Remarks to the Author):

Comments to Kulczyk et al.(NCOMMS-22-39611-T)

I have reviewed the previous version of this paper for Nature Struct Mol Biol and gave number of comments and suggestions. In this extensively revised version, the authors addressed most of my concerns and criticisms by performing additional experiments and re-analyses, adding many supplemental display items, and adding detailed description in the methods section. I am satisfied with these revisions and can confidently recommend acceptance of this paper for Nature Communications.

Jun Takagi

We are grateful to Professor Junichi Takagi for his constructive comments and criticism, which helped us to revise and improve the manuscript.